# Lipidation of Antimicrobial Peptides as a Design Strategy for Future Alternatives to Antibiotics

**DOI:** 10.3390/ijms21249692

**Published:** 2020-12-18

**Authors:** Taylor Rounds, Suzana K. Straus

**Affiliations:** Department of Chemistry, University of British Columbia, 2036 Main Mall, Vancouver, BC V6T 1Z1, Canada; trounds@uoguelph.ca

**Keywords:** antimicrobial peptide (AMP), host-defense peptide (HDP), lipidation

## Abstract

Multi-drug-resistant bacteria are becoming more prevalent, and treating these bacteria is becoming a global concern. One alternative approach to combat bacterial resistance is to use antimicrobial (AMPs) or host-defense peptides (HDPs) because they possess broad-spectrum activity, function in a variety of ways, and lead to minimal resistance. However, the therapeutic efficacy of HDPs is limited by a number of factors, including systemic toxicity, rapid degradation, and low bioavailability. One approach to circumvent these issues is to use lipidation, i.e., the attachment of one or more fatty acid chains to the amine groups of the N-terminus or a lysine residue of an HDP. In this review, we examined lipidated analogs of 66 different HDPs reported in the literature to determine: (i) whether there is a link between acyl chain length and antibacterial activity; (ii) whether the charge and (iii) the hydrophobicity of the HDP play a role; and (iv) whether acyl chain length and toxicity are related. Overall, the analysis suggests that lipidated HDPs with improved activity over the nonlipidated counterpart had acyl chain lengths of 8–12 carbons. Moreover, active lipidated peptides attached to short HDPs tended to have longer acyl chain lengths. Neither the charge of the parent HDP nor the percent hydrophobicity of the peptide had an apparent significant impact on the antibacterial activity. Finally, the relationship between acyl chain length and toxicity was difficult to determine due to the fact that toxicity is quantified in different ways. The impact of these trends, as well as combined strategies such as the incorporation of d- and non-natural amino acids or alternative approaches, will be discussed in light of how lipidation may play a role in the future development of antimicrobial peptide-based alternatives to current therapeutics.

## 1. Introduction

Due to the overuse or misuse of antibiotics, multi-drug-resistant bacteria are becoming more prevalent [1,2,3]. The ESKAPE pathogens (*Enterococcus faecium*, *Staphylococcus aureus*, *Klebsiella pneumoniae*, *Acinetobacter baumannii*, *Pseudomonas aeruginosa*, *and Enterobacter* spp.) are the leading cause of hospital-acquired infections [4], and thus their growing resistance to antibiotics is a global concern [4,5]. One alternative approach that is being explored is the use of antimicrobial peptides (AMPs) [6,7,8,9,10], which are also known as host-defense peptides (HDPs). These are small active peptides that are found in all life forms, including bacteria and plants [7]. AMPs are beneficial to bacteria because they are capable of killing other species of bacteria that may compete for nutrients. Likewise, HDPs play a key role in protecting plants against bacterial infection. Thus, AMPs have received significant attention because of their antimicrobial activity against a broad spectrum of pathogens, including both Gram-positive and Gram-negative bacteria [11,12,13,14]. In addition, HDPs attack bacteria using a mechanism that is different from that used by most conventional antibiotics, and as a result, little evidence of resistance has been seen against these peptides [6,7,13,15]. Conventional antibiotics typically act on a specific cellular target as a way of initiating cellular death [16]. However, many HDPs use nonmediated membrane destruction in which disruption of the cell membrane leads to cell lysis and ultimately death. As a result, it is more difficult for bacteria to develop resistance to HDPs. In addition to this, evidence has also shown that HDPs can enter a bacterial cell in order to interact with intracellular targets by binding to DNA and RNA or promoting the production of reactive oxygen species to cause further damage [8,16]. Finally, HDPs can additionally modulate the immune response [17,18,19] or display anticancer activity [20,21,22]. This diversity in their mode of action has sparked great interest in the clinical potential of these HDP-based therapies.

Despite their potential, HDPs have limitations due to a number of factors. Firstly, natural HDPs mostly consist of l-amino acids, making them susceptible to protease degradation and rapid kidney clearance [23,24,25]. Moreover, some AMPs display systemic toxicity by not being specific to bacteria. Indeed, systemic administration results in cytotoxic profiles in blood and short half-lives in vivo [17]. Many approaches have been investigated to circumvent these limitations [7,8]. These include the use of delivery vehicles [26] and chemical modification of AMPs [27]. Approaches to increase activity, specificity, biocompatibility, and stability include the inclusion of unnatural amino acids [27,28,29], peptide cyclization [30,31,32], the incorporation of d-amino acids [7,33,34], as well as the conjugation of carbohydrates [35] or fatty acids [36,37,38].

Lipidation, i.e., the attachment of one or more fatty acid chains to N-terminal residues or lysine side-chains of an HDP, has been of particular interest. It has been shown in previous studies that lipidation of AMPs can improve the antimicrobial potency of the peptides without altering their properties [36,39]. It has been suggested that this improved potency is due to enhanced interactions between the bacterial cell membrane and the fatty acid–peptide conjugate [38]. However, although conjugation of fatty acids to the peptide can cause improvements to antimicrobial activity, this is not always the case as the antimicrobial activity is highly dependent on the length of the acyl chain. As acyl chains increase in length, there is an increased tendency for self-assembly to occur. As a result of the altered water–membrane partition equilibrium, the hydrophobic chains conjugated to the peptides would have reduced interaction with water, and thus this reduces the chance of peptide–membrane interaction. In addition, as the length of the conjugated fatty acids increases, toxicity towards mammalian cells increases due to poor membrane selectivity [35,38]. Overall, although the acylation of AMPs has been proven to provide benefits to the antimicrobial activity, there appear to be optimal lengths of fatty acid chains in which the antimicrobial activity is increased, but the toxicity towards mammalian cells is not a concern. This literature review will provide an overview of the previous studies in which various lengths of fatty acid chains have been conjugated to AMPs, and the effects on antibacterial activity and toxicity will be discussed. The trends that are seen based on the hydrophobicity of the original peptide, the number of amino acids in the peptide, and the charge of the peptide will be studied to determine if any direction to future studies can be given. Basic design rules may provide important guidelines for the design of future lipidated HDP-based alternatives to antibiotics.

## 2. Review of the Lipidation Literature

Sixty-six different HDPs, which had been lipidated or modified in other ways (e.g., conjugation with benzoic acid [38], cinnamic acid [40], and so on), were found in papers dated from 1978 to 2020, with most references dating from 2002 onwards. Lists of minimum inhibitory concentrations (MIC) against a number of bacteria (*E. coli*, *P. aeruginosa*, *A. baumannii*, and *S. aureus*), minimum biofilm eradication concentrations (MBEC) against *S. aureus*, and toxicity were generated for these 66 HDPs. Peptides for which data with a range of acyl chain lengths were available were considered further (Table 1). Excluded HDPs include studies where the peptide sequence was varied, but not the length of the acyl chain [41], or studies where only the HDP + 1 acyl-modified HDP were reported [42]. For the remaining 47 HDPs, their length, charge (for the peptide alone), and percent hydrophobic content were tabulated and are given in Table 1.

### 2.1. Length of Acyl Chain

For each of the 47 different HDPs in Table 1, the number of lipidated HDPs that represented the most active antibacterial (i.e., lowest MIC) for a given acyl chain length were tallied and counted. Figure 1a shows that the optimal chain lengths are most commonly found to be C8–C12. Interestingly, this trend is consistent for *E. coli*, *P. aeruginosa*, and *S. aureus*. It should be noted that the trend is less clear for *A. baumanii*, most likely due to the fact that there were only 13 HDPs for which data were available, as compared to the >40 peptides for each of the other bacteria. The aim was to also see if this trend is mirrored in terms of MBEC values, but there were only five values available in our list, so we did not analyze MBEC data further. 

The trend shown in Figure 1a is possibly linked to these acyl chain lengths being most conducive to secondary structure formation and membrane insertion. Indeed, Mak et al. showed that acylated peptides, specifically with acyl chains of 8, 10, or 12 carbons, were observed to have higher *α*-helical content in comparison to the parent peptide in the presence of large unilamellar liposomes or trifluoroethanol, a membrane mimic [74]. Moreover, these same lipidated helical peptides inserted into the liposomes most effectively, demonstrating that the *α*-helical structure of a lipopeptide is clearly correlated to the ability to disrupt a bacterial membrane. Furthermore, conjugates with longer acyl chains may lead to a decrease in the activity due to aggregation and self-assembly [38,48,50,58]. Indeed, Sikorska et al. showed that the monomeric forms of short arginine-rich HDPs were responsible for the disruption of the bacterial membrane as opposed to aggregated forms [58]. In order to examine whether the length of the HDP plays a role in this trend, the data were re-examined in terms of the relative ratio of HDP length and acyl chain length.

### 2.2. Relative HDP and Acyl Chain Lengths

In Figure 1b, the average ratio of HDP length to acyl chain length is presented as a function of the acyl chain length. This takes into account all data for the four bacterial strains listed in Section 2.1. What can be observed is that for short chain lengths, the HDPs tend to be longer than for longer acyl chain lengths (bars, Figure 1b). Interestingly, the total length of HDP and acyl chain is fairly constant at 20 (number of amino acids + acyl Cs, squares and fitted line, Figure 1b). This translates to a molecule that would have a total length of 31–33 Å, assuming the HDP is *α*-helical, which most HDPs in Table 1 are. This number matches more closely the POPC/POPG bilayer thickness of ~39 Å than the individual peptide, suggesting that the lipidated peptide might insert more readily. POPC/POPG bilayers have been shown to be reasonable models to mimic bacterial membranes [75]. 

### 2.3. Charge of HDPs

Next, the effect of the charge of the HDP was examined to see if any trends were to be observed. Figure 1c shows the average charge of the HDPs as a function of acyl chain length, again using the data for all the bacterial strains discussed in Section 2.1. Here, there are no clear trends to be seen. This suggests that the burial of the acyl chain may be the driving force for the insertion of lipidated peptides, making cationicity less important.

### 2.4. Hydrophobic Content of HDPs

Finally, the effect of the hydrophobic content of the HDP was examined. Figure 1d shows the average hydrophobicity of the HDPs as a function of acyl chain length (using data from Section 2.1). As with charge, there are no clear trends to be seen, which could again imply that the hydrophobicity of the acyl chain is the important factor leading to the efficient insertion of lipidated peptides into the bacterial membrane.

### 2.5. Caveats

It should be noted that the number of data points for certain parameters is limited; e.g., *A. baumannii*, C18 HDPs. Consequently, the trends described above should be used as rough rules of thumb in designing lipidated HDPs. More detailed molecular descriptors [76,77,78] provide more stringent design rules.

### 2.6. Toxicity

An attempt to examine how toxicity depends on acyl chain length was also made. As observed by Kamysz et al. [38], such comparisons are, however, difficult to make because of the lack of uniformity in the reported toxicity values. Some studies list HC50 values [35], i.e., concentrations where 50% hemolysis occurs, whereas others report minimum hemolytic concentrations (MHC) [38]. Others still give the % of hemolysis relative to the MIC [39]. Furthermore, one general observation can be made on the dependence of toxicity on acyl chain length: the longer the acyl chain, the greater the hemolysis. For example, Figure 3 in [38] and in [35] clearly exemplify this. As mentioned above, this trend is likely due to poor membrane selectivity [35,38], i.e., the insertion of the lipidated peptide for long acyl chains is driven by the burial of hydrophobic groups and is insensitive to the surface composition of the membrane.

## 3. Combining Lipidation with Other Methods to Increase Biocompatibility, Stability, Activity, and Specificity 

A number of studies encountered during the literature review used a combination of approaches to improve the activity of lipidated HDPs. For example, Zhong et al. [16] found that exchanging two lysine l-amino acids for the d-enantiomers in anoplin significantly decreased the antibacterial activity of the peptide. However, the conjugation of acyl chains to the peptide containing d-enantiomers improved the antibacterial activity better than the parent peptide, and in addition, resulted in better stability against proteases. Another study conducted by Lee et al. on peptide CopW demonstrated that exchanging all l-amino acids for d-amino acids increased the antibacterial activity against some bacterial strains [65]. The antibacterial activity was further improved when acyl chains were conjugated to the N-terminus. The peptide composed of d-amino acids was also shown to have significantly better stability in serum. There has been much evidence to suggest that stability can be provided through the incorporation of d-amino acids into HDPs [66,79,80,81]. The addition of lipidation serves to further improve activity, making the resulting compounds of interest.

Although examples of lipidated HDPs with non-natural amino acids have been reported (Table 1) [59,82], to the best of our knowledge, there are no studies that systematically examine the effect of changing l-amino acids for non-natural ones on the activity of lipidated HDPs of varying acyl chain length. Since a number of studies have shown that incorporating unnatural amino acids can improve stability [27,28,80], it is expected that this combined approach could yield valuable antimicrobials. Indeed, a number of studies on lipidated peptoids have been reported [83,84] and show that the combined use of acyl chains and non-natural peptides are highly active.

## 4. Comparison of Lipidation to Other Methods to Increase Biocompatibility, Stability, Activity and Specificity

In order to assess how valid lipidation is as a design strategy for future alternatives to antibiotics, it is important to understand its advantages and disadvantages in the context of other currently available methods (Figure 2), i.e., chemical modification of HDPs such as the inclusion of d-amino acids, the rational substitution of amino acids, cyclization/stapling, and peptidomimetics. 

The substitution of l-amino acids for d-amino acids (Figure 2) is an approach typically used to prevent the degradation of HDPs by proteases [7]. Generally, changing the stereochemistry of an amino acid often does not significantly affect the antimicrobial activity of the peptide. For example, the MICs of the d-amino acid counterpart of peptide 73 were very similar to its fully l-amino acid version [79]. In other words, this approach is not usually used as a way to improve activity but rather simply to improve stability. For instance, Jia et al. replaced all l-amino acids for d-amino acids in the HDP polybia-CP, thereby making it resistant to trypsin and chymotrypsin degradation, without compromising its activity against both Gram-negative and Gram-positive bacteria [33]. Moreover, these authors noted that the single substitution of the l-lysine for d-lysine helped to improve stability and substantially decreased its hemolytic activity while only slightly decreasing antimicrobial activity [34]. Furthermore, studies have shown that introducing d-amino acids can lower host toxicity [79,85,86,87]. Finally, compared to lipidation, d-peptide synthesis is more costly [88], thereby often hindering its application in the clinic.

The rational substitution of amino acids to generate new HDPs has been explored extensively over the years [6,7,89,90]. For example, a recent review [91] has illustrated how peptides can be engineered from HDPs found in nature, e.g., temporins and aureins. This approach is typically used to improve antimicrobial activity and reduce hemolytic activity [88,92,93]. Sequences are varied by changing the position of positively charged amino acids (e.g., Lys) within the hydrophilic and hydrophobic segments of the HDP [88,92]. For example, antimicrobial activity against both Gram-positive and Gram-negative bacteria has been shown to improve by the substitution of lysine in the hydrophilic face of magainin II. This has been complemented with slight reductions in hemolytic activity. Conversely, activity is affected in various ways when Lys substitution occurs in the hydrophobic face of magainin, the extent of which depends on the residue being replaced [88]. Interestingly, the positioning of Lys in the hydrophobic face of the HDP is accompanied by a decrease in hemolysis. Alternatively, more active peptides are designed by adding arginine and/or tryptophan residues [89,94,95]. Cationic residues (i.e., Lys and Arg) help facilitate the initial electrostatic interaction between HDPs and the bacterial cytoplasmic membrane [90,95], whereas Trp residues facilitate peptide–lipid interactions (Figure 2) by preferably binding to the interfacial region of membranes [90]. In addition, the use of both Trp and Arg amino acids allows for cation-*π* interactions, which can further contribute to favorable peptide–membrane interactions [89,95]. Overall, the process of rational substitution serves to directly reduce MICs and hopefully also improve selectivity, as most approaches optimize activity first and foremost. Compared to the lipidation approach, the substitution of amino acids typically involves the generation of large libraries of peptides (e.g., >5–10 [91] to multiple hundreds of peptides [76,96]). In contrast, most of the lipidation studies reviewed in Section 2 involved the synthesis of <10 analogs. However, lipidation generally results in less selective compounds, as discussed in Section 2.5.

Cyclization is another common method to protect peptides against protease degradation. A number of approaches exist to accomplish this, such as head-to-tail cyclization, side-chain-to-tail cyclization, and using disulfide bridges via cysteine residues [97]. Alternatively, one or more staples (i.e., using *α*–methylated amino acids [98]) can be incorporated between *i*, *i* + 4 or *i* + 7 residues (Figure 2), thereby locking the peptide into an *α*-helical structure with twisted amide bonds, which makes them less favorable to proteolytic cleavage [30,88]. Alternatives to all hydrocarbon-based approaches include nitrogen and sulfur arylation of lysine and cysteine residues, respectively [31,32]. Though cyclization does indeed increase resistance to proteolysis, it often comes with unpredictable and undesired effects on mammalian membrane lysis, antimicrobial activity, and peptide solubility [30,88]. Recently, a systematic study by Mourtada et al. examined the structure–function–toxicity relationship of 58 stapled AMP (StAMP) constructs of magainin II in order to devise an algorithm to predict stable, nontoxic StAMPs [88]. The authors sequentially incorporated *i*, *i* + 4 staples at different positions in the magainin II sequence so as to determine which staple placement had minimal undesired effects on hemolytic activity. Interestingly, they found that when staples were placed within an already established hydrophobic face, the resulting StAMPs were less hemolytic. On the other hand, placement of stapled in a region of low hydrophobicity had an impact on the activity. Using their algorithm, Mourtada et al. were finally able to generate de novo StAMPs that showed strong antimicrobial activity and little to no hemolysis. This was done without having to resort to a costly synthetic library. Recent work by Etayash et al. [97] has shown that cyclization of the HDP IDR-1018 only slightly improved the MIC relative to the parent peptide. The main advantage in the cyclization of IDR-1018 was seen in the results obtained from a murine cutaneous abscess model, where the stability of the cyclic peptide was directly correlated to improved wound healing. Overall, the cyclization of peptides involves a number of complex steps, which can result in low yields [98]. Lipidation, on the other hand, appears to be more straightforward, as long as strategies such as a second coupling [35] are used.

Finally, peptidomimetic approaches involve using modified amino acids or amino-acid-like units (Figure 2) to generate mimics that imitate the structure, activity, and mode of action of HDPs. A key design feature is that the overall amphiphilic structure of the original peptide is maintained [7,99,100]. As a result of their non-natural composition, peptidomimetics are able to improve in vivo half-life and stability relative to HDPs. In many cases, the use of mimics also helps improve toxicity and lower synthesis costs [9,99,100,101]. Indeed, many successful HDPs in clinical trials are nonpeptidic derivatives, such as, e.g., the defensin mimetic brilacidin [9,102]. Other examples include the recent work by Luther et al., who recently developed a class of peptidomimetics built using a mixture of synthetic building blocks and natural amino acids arranged into two linked macrocycles [103]. A number of these derived compounds exhibited strong efficacy in a variety of mouse models of infection, activity against a wide array of bacteria (including in serum), favorable tolerability and pharmacokinetics, and low cell toxicity. A number of reviews can be consulted for further examples [9,99,100]. The complexity and cost of synthesis appear to be on par with lipidation approaches.

Overall, one of the main advantages of lipidation over the alternative strategies presented above appears to be the marked improvement in the activity of lipidated HDPs [35,37,38], which can be easily achieved through the N-terminus conjugation of C_8_–C_12_ fatty acid chains, using a small number of constructs. Moreover, the cost and ease of synthesis appear to be generally lower than some of the other methods. Although many of the studies surveyed here do not examine stability, some examples can be found in the literature. For instance, Lombardi et al. used the fact that lipidated HDPs tend to self-assemble to generate nanostructures that showed improved antibiofilm activity and protease stability [37]. These compounds were made by linking the HDPs to an aliphatic polyalanine peptide, which in turn was attached to a C_19_ lipidic tail. In terms of disadvantages, the main problem with lipidation is the increased cytotoxicity of the peptides, which increases with larger carbon length [35,38,104].

## 5. Discussion

The increase in levels of bacterial antibiotic resistance has led researchers to look for alternatives to current antibiotics [3,105]. HDPs or synthetic analogs are considered to be interesting substitutes because they can function in a variety of ways [6,7,106]. Recently, a number of studies have shown that lipidation results in HDPs with improved activity. As long as the chain length is well chosen, a balance exists between improved antibacterial properties and selectivity. Interestingly, many of the active lipidated peptides have acyl chain lengths of C_8_–C_12_ (Figure 1a) and short HDPs have longer acyl chains so that the overall length of the system matches closely the bacterial membrane bilayer thickness. The net charge of the HDP and its hydrophobic content play no significant role in determining activity. Most of the lipidated peptides surveyed here had the acyl chain conjugated at the N-terminus. However, there were a few studies in which the acyl chains were conjugated at locations that were not the N-terminus and these were successful as well. For example, the study by Albada et al. [46] attached acyl chains to lysine side-chains. Similarly, other examples have been reported in which acyl chains have been attached to the lysine side-chain and where improvements in antibacterial activity have been observed [35,55,67]. Another unique approach is shown in the study by Stachurski et al. [56], in which 2,4-diaminobutyric and 2,3-diaminopropionic acids are used as linkers at the N-terminus to which the acyl chains were attached.

Overall, designing future lipidated HDPs is a useful strategy for developing viable alternatives to antibiotics. A comparison with alternative methods to improve biocompatibility, stability, activity, and specificity demonstrated that lipidation offers many advantages. Although toxicity is still an important impediment to the wide use of lipidated HDPs, future designs may find a way to circumvent this issue. Furthermore, each new design will provide a better fundamental understanding of the important features required for activity and minimal toxicity. Finally, a combination of approaches, i.e., using d-amino acids or unnatural amino acids, cyclization, and so on, will ensure that lipidated HDPs become viable future alternatives to antibiotics.

## Figures and Tables

**Figure 1 ijms-21-09692-f001:**
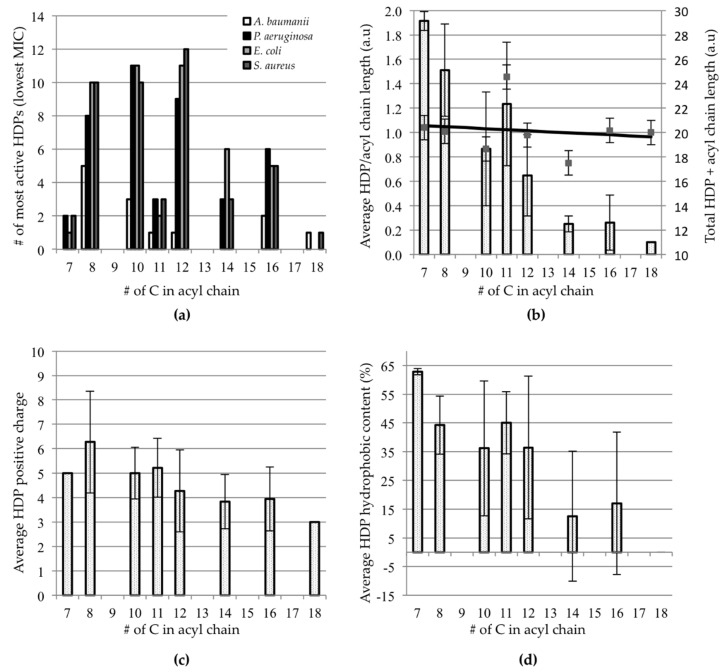
Comparison of lipidated HDPs listed in Table 1 as a function of acyl chain length. (**a**) Number of lipidated HDPs for which a given acyl chain length results in the lowest MIC for 4 different bacteria; (**b**) average ratio of HDP length and acyl chain length (number of amino acids/number of Cs in the acyl chain)—bars; and total length (squares and fitted line); (**c**) average charge of the parent (nonlipidated) HDP; (**d**) average percent hydrophobic content for the parent HDP. In panel (**b**), a comparison of C8 versus C12 and C8 versus C16 yields *p*-values < 0.001. For panels (**c**,**d**), a similar comparison yields p-values that indicate that the differences are not significant.

**Figure 2 ijms-21-09692-f002:**
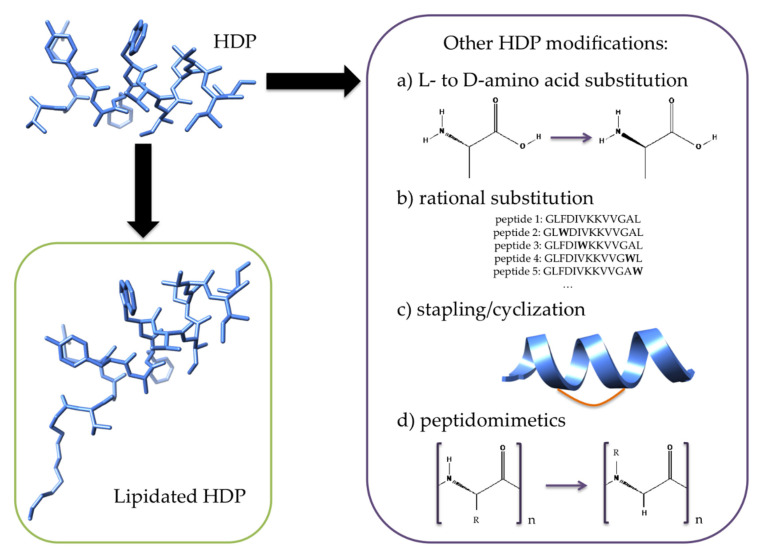
Comparison of lipidated HDPs (green box) to alternative approaches (purple box): replacement of one or more l-amino acids for d-amino acids in the HDP, rational substitution of amino acids, stapling or cyclization, and, finally, the use of peptidomimetics, where, e.g., the side chain (designated as R) is not on the C*α* but rather on the N atom.

**Table 1 ijms-21-09692-t001:** List of host-defense peptides (HDPs) and some of their properties used to generate the plots in Figure 1.

Peptide	Sequence	# Amino Acids	Charge ^1^	% Hydrophobic ^2^	Reference
1	KRIVQRIKDFLR	12	+5	42	[38]
2	FQWQRNIRKVR	11	+5	36	[43]
3	OOWW ^3^	4	+3	50	[40,44,45]
4	RWRWRW	6	+4	50	[46]
5	LWKTLLKKVLKAAA	14	+5	64	[47]
6	WKTLLKKVLKAAA	13	+5	62	[47]
7	KTLLKKVLKAAA	12	+5	58	[47]
8	TLLKKVLKAAA	11	+4	64	[47]
9	ALWKTLLKKVLKA	13	+5	62	[48]
10	KYR	3	+3	0	[49]
11	SKVWRHWRRFWHRAHRKL	18	+8	39	[50]
12	YGAAKKAAKAAKKAAKAA	18	+7	56	[42,51]
13	KLLK	4	+3	50	[52]
14	KAAK	4	+3	50	[52]
15	KKK	3	+4	0	[53]
16	RRWQWRMKK	9	+6	33	[54]
17	LKKLLKKLLKKL	12	+7	50	[39]
18	KKKK	4	+5	0	[55]
19	DapKKK ^4^	4	+5	0	[56]
20	DabKKK ^5^	4	+5	0	[56]
21	KKLLKLLLKLLK	12	+6	58	[57]
22	GIGKFLHSAKKWGKAFVGEIMNS	23	+4	43	[57]
23	RRRR	4	+5	0	[58]
24	VDabGSWSDabDabFEVIA	13	+3	46	[59,60]
25	RGRKVVRRKK	10	+8	20	[61]
26	RGRKGGRRKK	10	+8	0	[61]
27	GATAIKQVKKLFKKKGG	17	+7	35	[62]
28	RWKRHISEQLRRRDRLQRQAJ	21	+7	25	[63]
29	IKQVKKLFKK	10	+6	40	[64]
30	LLWIALRKK	9	+4	67	[65]
31	KKLLKLLLKLLK	12	+6	58	[66]
32	KKLLKKLKKLLK	12	+8	42	[66]
33	KKKLKKLKKKLK	12	+10	25	[66]
34	GLLKRIKTLL	10	+4	50	[16,67]
35	RKWWK	5	+4	40	[68]
36	RFWR	4	+3	50	[69]
37	RR	2	+3	0	[70]
38	KKC	3	+3	0	[71]
39	KR	2	+3	0	[71]
40	O	1	+2	0	[72]
41	OO	2	+3	0	[72]
42	OOO	3	+4	0	[72]
43	OOOO	4	+5	0	[72]
44	OOOOO	5	+6	0	[72]
45	RIRIRWIIR	9	+5	56	[35]
46	KRRVRWIIW	9	+5	56	[35]
47	YVLWKRKRKFCFI	13	+6	46	[73]
	AVERAGE ± St. Deviation	9 ± 5	5 ± 2	32 ± 24	

^1^ Charge for nonlipidated HDPs. Most HDPs have an amidated C-terminus. ^2^ Using https://www.peptide2.com/N_peptide_hydrophobicity_hydrophilicity.php for peptides with natural amino acids. ^3^ O = Orn. ^4^ Dap = 2,3-diamino-propionic acid. ^5^ Dab = 2,4-diaminobutyric acid.

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
