# Peer review of "Lipidation of Antimicrobial Peptides as a Design Strategy for Future Alternatives to Antibiotics"

_ijms, 2020, doi:10.3390/ijms21249692_

Round 1
Reviewer 1 Report
In this review, the authors examined lipidated analogues of 66 different HDPs reported in the literature to determine: i) whether there is a link between acyl chain length and antibacterial activity; ii) whether the charge and iii) the hydrophobicity of the HDP play a role. Overall, the analysis suggests that lipidated HDPs with improved activity over the non-lipidated counterpart had acyl chain lengths of 8-12 carbons. Moreover, active lipidated peptides attached to short HDPs tended to have longer acyl chain lengths. Finally, neither the charge of the parent HDP nor the percent hydrophobicity of the peptide had an apparent significant impact on the antibacterial activity. In my opinion, this review is well-written and is appropriate for publication in IJMS in its current form.
Author Response
We thank the reviewer for their positive comments.
Reviewer 2 Report
Authors did a nice job collecting all this information into a comprehensive unit. I would like to see some additional info about the Toxicity and this chapter should be significantly extended with some graphical representations.
Author Response
We thank the reviewer for their positive comments and have addressed their points in the following way:
1) Toxicity: there was a section already in the manuscript on this and it would be difficult for us to add more as a comparison across all 66 peptides is not possible because different authors report different toxicity numbers.
We did change the abstract though to make sure that toxicity is mentioned there, as it was not before. Perhaps this was the reason for the reviewer's comment.
2) Additional graphical representations: We have added a new figure (Figure 2) which summarizes the alternative methods discussed in Section 4.
We thank the reviewer for their insightful comments, which have helped to improve the manuscript. We hope that we have addressed all concerns in a satisfactory manner.